# How far from the gold standard? Comparing the accuracy of a Local Position Measurement (LPM) system and a 15 Hz GPS to a laser for measuring acceleration and running speed during team sports

**Karin Fischer-Sonderegger**[1,2]*, **Wolfgang Taube**[2], **Martin Rumo**[1], **Markus Tschopp**[1]

1 Swiss Federal Institute of Sport Magglingen SFISM, Section for Elite Sport, Magglingen, Switzerland,
2 Department of Neurosciences and Movement Science, University of Fribourg, Fribourg, Switzerland

* karin.fischer@baspo.admin.ch

## Abstract

### Purpose

This study compared the validity and inter- and intra-unit reliability of local (LPM) and global (GPS) position measurement systems for measuring acceleration during team sports.

### Methods

Devices were attached to a remote-controlled car and validated against a laser. Mean percentage biases (MPBs) of maximal acceleration ($a_{max}$) and maximal running speed ($v_{max}$) were used to measure validity. Mean between-device and mean within-device standard deviations of the percentage biases (bd-SDs and wd-SDs) of $a_{max}$ and $v_{max}$ were used to measure inter- and intra-unit reliability, respectively.

### Results

Both systems tended to underestimate $a_{max}$ similarly (GPS: −61.8 to 3.5%; LPM: −53.9 to 9.6%). The MPBs of $a_{max}$ were lower in trials with unidirectional linear movements (GPS: −18.8 to 3.5%; LPM: −11.2 to 9.6%) than in trials with changes of direction (CODs; GPS: −61.8 to −21.1%; LPM: −53.9 to −35.3%). The MPBs of $v_{max}$ (GPS: −3.3 to −1.0%; LPM: −12.4 to 1.5%) were lower than those of $a_{max}$. The bd-SDs and the wd-SDs of $a_{max}$ were similar for both systems (bd-SDs: GPS: 2.8 to 12.0%; LPM 3.7 to 15.3%; wd-SDs: GPS: 3.7 to 28.4%; LPM: 5.3 to 27.2%), whereas GPS showed better bd-SDs of $v_{max}$ than LPM.

### Conclusion

The accuracy depended strongly on the type of action measured, with CODs displaying particularly poor validity, indicating a challenge for quantifying training loads in team sports.

**Data Availability Statement:** All relevant data are within the manuscript and its Supporting information files.

**Funding:** The authors received no specific funding for this work.

**Competing interests:** The authors have declared that no competing interests exist.

# Introduction

Team sports are characterized by frequent changes of direction (CODs) and many accelerations and decelerations [1, 2]. Since accelerating consumes more energy than maintaining a constant speed, measuring the distance achieved at different speeds does not completely reflect training or game loads [1, 3–5]. Therefore, in order to assess short, intense actions that are typical of team sports, position measurement systems that can accurately measure acceleration are needed [1]. Both global (GPS) and local (LPM) position measurement systems have been used to this end [6–8]. GPS uses satellites to determine the positions of players; it is therefore not locally bound and can be used flexibly across different sites (e.g., home and away games, training and match fields). However, the number of available satellites can influence GPS's accuracy [9]. Satellites may also be hidden or have their signals reflected by physical obstacles such as buildings or trees, and can furthermore be affected by atmospheric conditions. In contrast, the LPM system uses base stations located around a specific site [7]. Its use is therefore limited to a fixed location, but is not affected by the satellite limitations of GPS and has a higher sampling frequency. We therefore hypothesized that the LPM measurement accuracy would be higher than GPS, particularly when measuring athletes' movement patterns in team sports.

Despite the importance of measuring accelerations to assess physical load, most studies that have sought to validate position measurement systems in team sports have compared running distances [6, 7, 10–12] or averaged speed [8, 11, 13, 14] to a gold standard. For total distance, the measurement error of three different 1 Hz GPS devices was established to be < 5% [6] and the coefficient of variation (CV) for 10 Hz and 18 Hz GPS ranged from 2.5 to 13.0% and 1.1 to 5.1%, respectively [10]. Similarly, Frencken et al. [7] used total distance in a team sport context to validate an LPM system's accuracy and found that it generally underestimated actual distance by up to –1.6% and mean speed by –0.1 to −0.6 km·h$^{-1}$. However despite fairly low error indicators for both systems, the key movements of team sports, such as accelerations and changes of direction, cannot be captured by such measurements [15]. Previous studies have accounted for accelerations when validating the accuracy of position measurement systems. Akenhead et al. [16] demonstrated that the validity and reliability of speeds measured with a 10 Hz GPS were inversely correlated to acceleration—i.e., the higher the acceleration, the lower the validity and reliability of speed measurements during the acceleration. Furthermore, Buchheit et al. [17] compared three different GPS models and showed a between-unit variation in peak acceleration of 10%; and the CV for maximal speed and maximal acceleration in Lacome et al. [18] was 0.5% and 6.4% respectively, as measured during a 40 m sprint test. However, to our knowledge, no studies have investigated accelerations during changes of direction.

While distances can be validated with a trundle wheel or measuring tape and averaged speeds can be measured with timing gates [6, 19, 20], there is no "gold standard" to assess acceleration in team sport-specific actions. Ideally, such a system should be able to measure both speed and acceleration, since the ability to accelerate depends on the speed at baseline. More specifically, maximal voluntary acceleration has been shown to decrease with increased initial running speed [21]. Lasers and radar beams are therefore more suitable measurement tools than timing gates [19, 22–24]. When used with adequate signal filtering, lasers are understood to accurately measure speed and acceleration [25], with an average speed error of < 2%, as reported by Tuerk-Noack and Schmalz [26]. Another study found that the typical error when using lasers to measure speed in a repeated running trial was very small (0.05 m·s$^{-1}$) and that the intra-class correlation was high ($r = 0.98$) [27]. However, it is also known that laser measurements' accuracy is limited during the first acceleration phase of a sprint due to shifts in the human body's center of gravity [28]. Furthermore, an athlete's upper-body movements

during a sprint can negatively influence the accuracy of the laser [16, 29, 30]. These limitations can be circumvented if the movement of the human body is simulated by a rigid body, such as a vehicle. Additionally, a vehicle offers the opportunity to wear several devices at the time. However, to date, only three GPS accuracy studies have analyzed acceleration using an object other than a person as a device carrier [16–18].

To determine the inter-unit reliability of several GPS and LPM devices, and less so the intra-unit reliability, a number of devices should be attached to the same person or object. Since there is limited space to adequately position multiple measurement devices on the back of a human, most studies to date have only used two to four devices to avoid encountering problems (e.g., discomfort, movement restrictions, additional weight) [11, 24, 31]. Other studies have approached this problem by attaching only one device at a time to an athlete's back and then instructing the athlete to complete a course several times [13, 32, 33]; however, this approach cannot differentiate between measurement errors of the different devices versus changes in the athlete's movement execution.

So far, no studies have assessed positioning system accuracy using a non-human object as a device carrier, while simultaneously comparing the accuracy of two different systems (i.e., global and local positioning systems) to a gold standard in different team sport-specific actions. Thus, the aim of this study was to determine and compare the validity of commercially available GPS and LPM systems in assessing acceleration and speed during forward and backward and single directional actions, using a laser measurement system as gold standard. Furthermore, this study aimed to assess the inter-unit and intra-unit reliability of GPS and LPM speed and acceleration measurements in different team sport-specific actions.

## Materials and methods

### Testing procedures

To examine the validity and the inter- and intra-unit reliability of a 15 Hz GPS and an LPM system, a remote-controlled car (RCC) (Traxxas Rally rushless 4WD 1/10 RTR, Model 70, Plano, Texas), steered by an operator, was used to carry the position measurement devices and simulate movement patterns common to team sports. One team sport-specific movement pattern can provide several team sport-specific actions (e.g. a movement pattern with two accelerations interspersed with an abrupt deceleration can be accounted for both low acceleration from standstill [LA] and acceleration after an abrupt deceleration [A-D]). Team sport-specific actions differing in acceleration capacity, initial speed, maximal speed, number of CODs, and were divided into seven subcategories. In order to present a number of trials within each subcategory (Table 1), sport-specific movement patterns were carried out several times. Throughout, the operator attempted to achieve speeds and accelerations that mimicked team sport-specific movement patterns as closely as possible. Accelerations occurred between 0 and 7.8

Table 1. Subcategories of different team sport-specific actions and the number of trials within the subcategory.

| Subcategory | # of trials |
| --- | --- |
| low acceleration from standstill (LA) | 8 |
| high acceleration from standstill (HA) | 6 |
| high acceleration from a flying start (HA-flyingS) | 6 |
| acceleration after an abrupt deceleration (A-D) | 4 |
| acceleration after a 180˚ change of direction (A-COD) | 4 |
| repetitive high acceleration, shuttle runs 4 x 5m (RA-5m) | 6 |
| repetitive high acceleration, shuttle run 4 x 10m (RA-10m) | 6 |

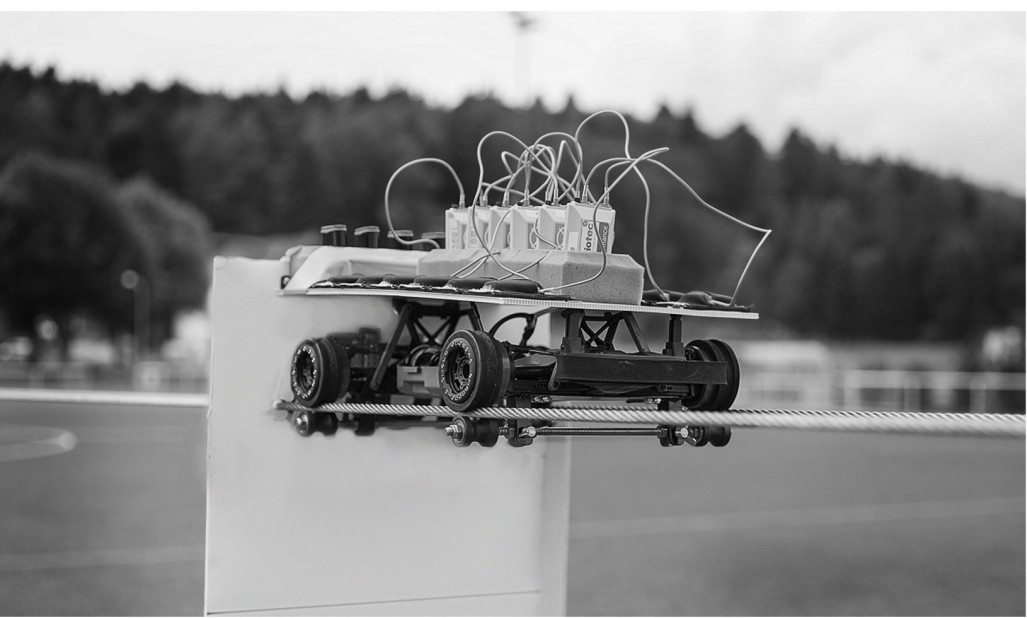

**Fig 1. Remote-controlled car (RCC) driving on two steel ropes.** Six LPM and six GPS devices were attached on the top of the RCC. At the back of the RCC was a reflective, white panel used to detect the laser beam.

$m \cdot s^{-2}$, and were classed as low acceleration if $< 3.5 \text{ m} \cdot s^{-2}$ or high acceleration if $> 3.5 \text{ m} \cdot s^{-2}$. Flying start accelerations were initialized from speeds between 6 and 15 $\text{km} \cdot h^{-1}$. This approach allowed us to identify differences in validity and inter- and intra-unit reliability of maximal speed ($v_{max}$) and maximal acceleration ($a_{max}$) during different types of actions.

Six GPS devices (15 Hz GPS, SPI HPU, GPSports Pty Ltd, Canberra, Australia) and six LPM devices (Inmotiotec GmbH, Regau, Austria) were simultaneously mounted in an upright position on a platform installed on top of the RCC. Care was taken to ensure that the LPM systems' antennas were horizontal, as they are when attached to an athlete's shoulder, and that all antennas were free from obstructions. The RCC's wheels were modified to allow it to drive on two steel ropes stretched 40 cm apart and 1.50 m above the ground, simulating the height at which position measurement devices are normally worn when attached to athletes (Fig 1). The steel ropes were stretched across the entire length of the soccer field through the middle of the field. The actions were carried out in the middle of the field.

The raw GPS data was interpolated by the device's software from 5 Hz to 15 Hz, as in Nagahara et al. [34]. An LPM system can record position data up to 1,000 Hz, but the effective frequency for each device is divided by the number of devices that are used. To reach the same frequency as in official games, 16 additional devices were randomly placed across the field, for a total of 22 devices. Thus, each LPM device recorded at a sampling frequency of 45 Hz (1,000 / 22). No trees or buildings were positioned around the measurement site, and testing was conducted under a clear blue sky with no cloud cover.

All collected GPS and LPM data were downloaded using customized software (SPI: Team AMS51; GPSports Systems, Canberra, Australia; LPM: Inmotio software, Inmotiotec GmbH, Regau, Austria) and then exported to Microsoft Excel (2016). The GPS and LPM customized software programs both provided speed and acceleration data by default. Validity and inter- and intra-unit reliability of speed and acceleration were tested by concentrating on $v_{max}$ and $a_{max}$ from each trial within a subcategory, which were manually extracted in a post-processing step.

A laser (LDM301, Jenoptic, Jena, Germany) with a sampling frequency of 100 Hz was used as the criterion to measure speed and acceleration of the RCC during both forward and backward and single directional movement patterns. The use of the RCC favored to ensure a complete laser signal, as a player's upper body can quickly shift outside the laser's range during a sprint compared to the car, which drives in a straight line with no lateral deviations. As a laser measures the time delay of a reflected pulsed infrared light, the sighting beam was focused on the center of a 0.3 m x 0.5 m white reflective panel secured on the RCC; the laser was located on a tripod 3m behind the start. Raw data was collected using the respective manufacturer software and then exported to Microsoft Excel (2016). During data post-processing, the first and second derivatives of the position data from the laser were calculated to obtain the speed and acceleration data. Subsequently, a moving average filter was used over 20 data points (forward and backward) to smooth the speed and acceleration data.

## Statistical analysis

The $v_{max}$ and $a_{max}$ means and between-device standard deviations (SDs) measured by GPS and LPM devices, as well as the $v_{max}$ and $a_{max}$ means measured by the laser, were assessed for each subcategory of the team sport-specific actions (Table 1). To evaluate the validity of the two position measurement systems, the mean percentage biases (MPBs) of $a_{max}$ and $v_{max}$ for GPS and LPM were expressed relative to the laser (gold standard), using the following formula: ($100 \times$ GPS parameter / laser parameter– 100) and ($100 \times$ LPM parameter / laser parameter– 100) (similar to Nagahara et al. [34]). The MPBs were calculated for each trial and averaged for each subcategory.

Similarly, to assess inter-unit reliability, between-device SDs of the percentage biases of $a_{max}$ and $v_{max}$ for all six devices were calculated for each trial and averaged for each subcategory. Additionally, the typical error was calculated and expressed as a CV.

For intra-unit reliability, the within-device SD of the percentage biases of $a_{max}$ and $v_{max}$ for each individual device was calculated over all trials and averaged for each subcategory. Both the between- and within-device SDs of the percentage biases, representing the inter- and intra-unit reliability, respectively, were expressed as a percentage relative to the laser-derived values. This approach was chosen because of the inevitable variations in the levels of acceleration and maximum speed between different trials within each subcategory. Using the SD of the absolute value instead of the SD of the percentage bias—especially when determining the intra-unit reliability—would have led to a misinterpretation of the reliability, due to the RCC's varying acceleration and speed curves from trial to trial.

**Table 2. Means and between-device standard deviations (SDs) of maximal acceleration ($a_{max}$) during different team sport-specific actions measured with a GPS and an LPM system and the mean of $a_{max}$ measured with laser.**

| Subcategory | Laser (m·s$^{-2}$) | GPS | | | LPM | | |
|---|---|---|---|---|---|---|---|
| | | Mean (m·s$^{-2}$) | SD (m·s$^{-2}$) | CV (%) | Mean (m·s$^{-2}$) | SD (m·s$^{-2}$) | CV (%) |
| low acceleration from standstill (LA) | 1.76 | 1.32 | 0.22 | 16.1 | 1.49 | 0.22 | 14.7 |
| high acceleration from standstill (HA) | 6.25 | 6.55 | 0.79 | 10.3 | 4.90 | 0.24 | 4.8 |
| high acceleration from a flying start (HA-flyingS) | 4.62 | 4.54 | 0.13 | 2.8 | 5.06 | 0.32 | 6.3 |
| acceleration after an abrupt deceleration (A-D) | 2.85 | 2.85 | 0.13 | 5.0 | 2.99 | 0.25 | 11.1 |
| acceleration after a 180˚ change of direction (A-COD) | 3.46 | 1.33 | 0.18 | 12.8 | 1.65 | 0.14 | 8.7 |
| repetitive high acceleration, shuttle runs 4 x 5m (RA-5m) | 5.08 | 3.04 | 0.29 | 9.8 | 2.94 | 0.32 | 10.7 |
| repetitive high acceleration, shuttle runs 4 x 10m (RA-10m) | 5.16 | 4.25 | 0.16 | 4.4 | 3.44 | 0.27 | 9.3 |

The Coefficient of Variation (CV) is an indicator of the inter-unit reliability.

## Results

### Validity of maximal acceleration

Table 2 shows the means and between-device SDs of $a_{max}$ during different team sport-specific actions as measured by the GPS and LPM systems, and the mean of the laser measurement. Fig 2A shows the MPB of $a_{max}$ as an indicator of validity. Both the GPS and the LPM devices tended to underestimate $a_{max}$, especially for actions that included CODs. The exceptions were high acceleration from standstill (HA) and A-D, which were slightly overestimated by GPS (MPB: 3.5% and 1.1%, respectively) and high acceleration from a flying start (HA-flyingS), which was overestimated by LPM (MPB: 9.6%). Moreover, the MPBs for $a_{max}$ were lower for both systems when measuring linear movements (LA, HA, HA-flyingS, and A-D) as compared to actions with CODs (acceleration after a 180˚ change of direction [A-COD], repetitive high acceleration, shuttle runs 4 x 5m [RA-5m], and repetitive high acceleration, shuttle runs 4 x 10m [RA-10m]) and were higher for RA-5m than for RA-10m. The highest MPB was in A-COD (GPS: −61.8%; LPM: −53.9%).

### Validity of maximal running speed

Table 3 shows the means and between-device SDs of $a_{max}$ during different team sport-specific actions as measured by the GPS and LPM systems, and the mean of the laser measurement.

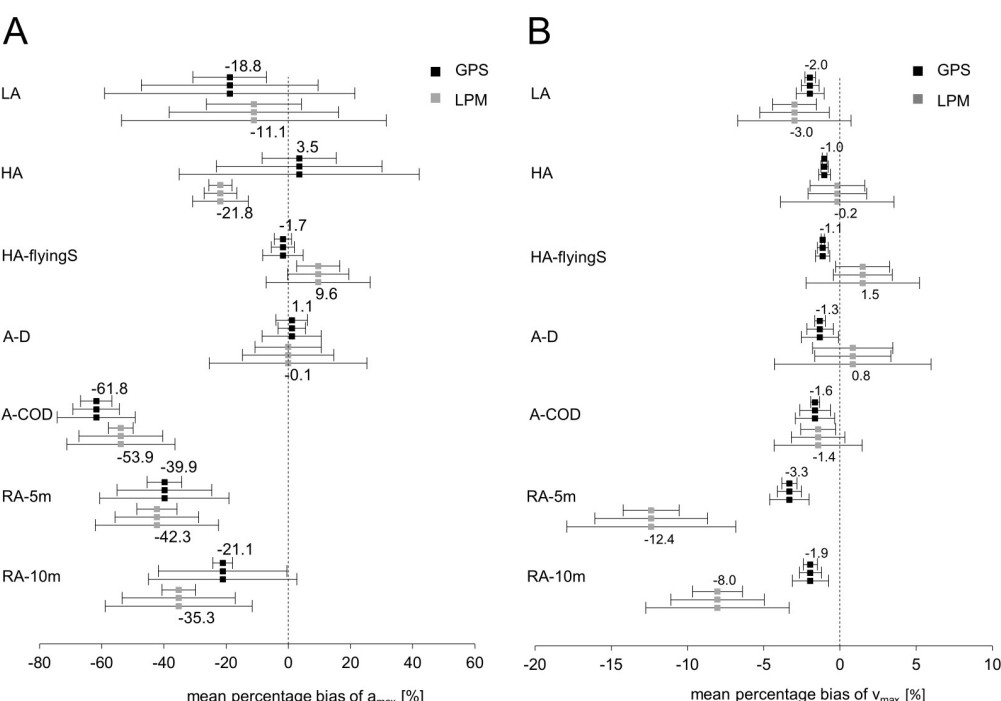

**Fig 2. Validity, inter- and intra-unit reliability of maximal acceleration ($a_{max}$; A) and maximal speed ($v_{max}$; B).** Mean percentage bias (MPB; relative bias of the GPS and LPM devices compared to the laser) of $a_{max}$ and $v_{max}$ of 7 different team sport-specific actions indicates the validity of $a_{max}$ and $v_{max}$. The first whisker of each subcategory within the black and grey squares is the between-device SD of the percentage biases of $a_{max}$ and $v_{max}$, and represents the inter-unit reliability of $a_{max}$ and $v_{max}$. The second whisker is the within-device SD of the percentage biases of $a_{max}$ and $v_{max}$, and represents the intra-unit reliability of $a_{max}$ and $v_{max}$. The third whisker of each subcategory is the combination of the between- and within-device SD of the percentage biases of $a_{max}$ and $v_{max}$, which indicates the inter- and intra-unit reliability of $a_{max}$ and $v_{max}$. If a GPS or LPM device is randomly chosen for repeated measures, the third whisker illustrates the percentage measurement error.

**Table 3. Means and between-device standard deviations (SDs) of maximal running speed ($v_{max}$) during different team sport-specific actions measured with a GPS and an LPM system and the mean of $a_{max}$ measured with laser.**

| Subcategory | Laser (m·s$^{-1}$) | GPS | | | LPM | | |
|---|---|---|---|---|---|---|---|
| | | Mean (m·s$^{-1}$) | SD (m·s$^{-1}$) | CV (%) | Mean (m·s$^{-1}$) | SD (m·s$^{-1}$) | CV (%) |
| low acceleration from standstill (LA) | 6.06 | 5.95 | 0.02 | 0.36 | 5.86 | 0.08 | 1.48 |
| high acceleration from standstill (HA) | 10.05 | 9.95 | 0.01 | 0.63 | 10.04 | 0.19 | 1.78 |
| high acceleration from a flying start (HA-flyingS) | 11.04 | 10.92 | 0.01 | 0.11 | 11.21 | 0.20 | 1.76 |
| acceleration after an abrupt deceleration (A-D) | 7.21 | 7.13 | 0.02 | 0.36 | 7.31 | 0.19 | 2.62 |
| acceleration after a 180˚ change of direction (A-COD) | 5.42 | 5.34 | 0.01 | 0.30 | 5.33 | 0.06 | 1.15 |
| repetitive high acceleration, shuttle runs 4 x 5m (RA-5m) | 4.79 | 4.63 | 0.02 | 0.51 | 4.20 | 0.09 | 5.44 |
| repetitive high acceleration, shuttle runs 4 x 10m (RA-10m) | 7.05 | 6.91 | 0.03 | 0.47 | 6.46 | 0.12 | 3.25 |

The Coefficient of Variation (CV) is an indicator of the inter-unit reliability.

The MPB of $v_{max}$ is presented in Fig 2B as an indicator of validity. In general, the MPBs of $v_{max}$ were markedly lower than of $a_{max}$; similar to $a_{max}$, $v_{max}$ tended to be systematically underestimated by GPS (−3.3 to −1.0%), whereas both negative and positive biases occurred with LPM (−12.4 to 1.5%). The largest differences between the GPS and LPM measurements were in RA-5m (GPS: −3.3%; LPM: −12.4%) and RA-10m (GPS: −1.9%; LPM: −8.0%). As shown in Fig 3, the LPM speed curve was clearly time-delayed during these repetitive accelerations, and its maximum recorded speed was considerably lower than the speed measured with the laser and GPS (with the exception of one LPM device in the second and fourth CODs).

## Inter- and intra-unit reliability of maximal acceleration

The inter-unit and intra-unit reliability of $a_{max}$, as measured by both GPS and LPM, is shown in Fig 2A as the between-device and within-device SDs of the percentage biases. Table 2 shows the CV of the GPS and LPM measurements, which can be also interpreted as an indicator of inter-unit reliability.

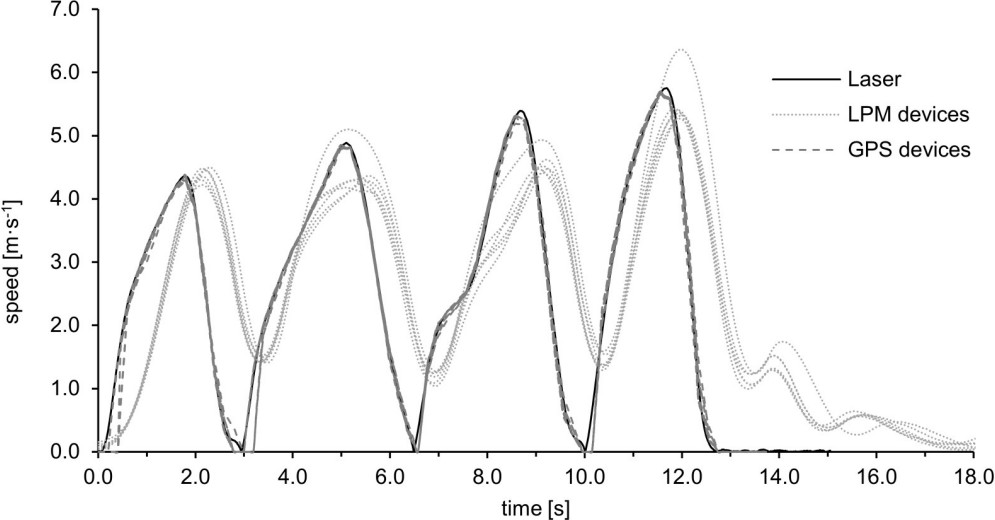

**Fig 3. Speed curve of the laser signal, 6 GPS and 6 LPM signals during repetitive high-accelerated shuttle runs 4 x 5m (RA-5m).**

The inter-unit reliability of $a_{max}$, as measured by GPS, ranged from 2.8 to 5.6% (expressed as the between-device SDs) for most of the tested actions, with the exceptions of LA (11.8%) and HA (12.0%). For LPM, the inter-unit reliability of $a_{max}$ ranged from 4.0 to 7.0%, with the exceptions of A-D (10.7%) and LA (14.6%).

The intra-unit reliability of $a_{max}$, as measured by GPS, ranged from 3.7 to 28.4% (expressed as the within-device SDs). Accelerations initiated from a standstill (HA and LA) had the lowest intra-unit reliability (26.6% and 28.4%, respectively), while HA-flyingS and A-D had the highest (3.7% and 4.4%, respectively). For LPM, the intra-unit reliability of $a_{max}$ ranged from 5.3 to 18.2%, with the exception of LA (28.3%); HA and HA-flyingS had the highest intra-unit reliability (5.3% and 9.8%, respectively).

For GPS, the highest inter- and intra-unit reliability of $a_{max}$—i.e., its summarized reliability —was found for HA-flyingS and A-D (6.5% and 9.5%, respectively), while HA and LA (38.6% and 40.3%, respectively) displayed the lowest summarized reliability. For LPM, HA generated the highest summarized reliability (9.0%) and LA the lowest (43.0%). The summarized reliability illustrates the percentage measurement error if a GPS or LPM device is randomly chosen for repeated measures.

### Inter- and intra-unit reliability of maximal running speed

Fig 2B shows the inter- and intra-unit reliability of $v_{max}$ measured by both GPS and LPM as between-device and within-device SDs of the percentage biases.

The overall inter- and intra-unit reliability of $v_{max}$ was considerably higher than that of $a_{max}$. The inter-unit reliability values for $v_{max}$ were all smaller than 0.5% when measured by GPS and smaller than 2.6% when measured by LPM, across all subcategories.

Furthermore, the *intra*-unit reliability of $v_{max}$ was lower than the *inter*-unit reliability for both systems, with the sole exception of A-D when measured by LPM. The intra-unit reliability was lower using LPM than GPS (GPS: between 0.2% and 1.0% and LPM: between 1.8% and 3.7%). Moreover, the GPS summarized reliability values for $v_{max}$ were below 1.3% across all subcategories, while the LPM summarized values for $v_{max}$ ranged from 2.9 to 5.6%.

## Discussion

The present study examined validity and inter- and intra-unit reliability of two different position measurement systems (local and global), when evaluating the accuracy of acceleration measurements in different team sport-specific actions. In contrast to our hypothesis, the current data revealed that in most cases the validity of $a_{max}$ in team sport-specific actions was not lower when measured with the 15 Hz GPS than with the LPM system. However, both systems' $a_{max}$ results demonstrated surprisingly large discrepancies compared to the "gold standard" laser measurements. On the other hand, the validity of $v_{max}$ was generally good and comparable to the "gold standard" in both systems.

Inter-unit reliability of $a_{max}$ was similar across both systems, while inter-unit reliability of $v_{max}$ was noticeably better with the GPS than with the LPM system. The inter-unit reliability of $a_{max}$ and $v_{max}$ was generally better than the intra-unit reliability using both measurement systems.

### Validity of the acceleration assessment

Overall, the results show that the validity for assessing $a_{max}$ using both systems is lower than the validity for assessing $v_{max}$. Although acceleration is the derivative of speed over time and it could be assumed that the $a_{max}$ biases would correlate to the $v_{max}$ biases, it must be noted that the $a_{max}$ values were attained earlier than the $v_{max}$ values in every action [21]. For this reason,

the $a_{max}$ measurement errors did not correlate with the $v_{max}$ ones. Furthermore, validity of $a_{max}$ (and $v_{max}$) not only depended on the type of measurement system, but also on the type of action being evaluated: both systems demonstrated the lowest validity of $a_{max}$ during actions with one COD (A-COD), and validity remained low in actions with multiple CODs (RA-5m, RA-10m). Furthermore, validity of $a_{max}$ was higher when high-accelerated actions are initiated from a flying start (HA-flyingS) than from standstill (LA, HA). The reason for the great difference in validity of $a_{max}$ between accelerations from standstill and from a flying start may be due to manufacturer's filter settings which smoothed the raw data [35, 36]. The Kalman filter explicitly relies on predictions of the next measurement and therefore assumes predictable trajectories. However, keeping trajectories unpredictable is an important factor in team sports. As shown in Fischer-Sonderegger et al. [37], soccer players are often already in motion before high acceleration occurs, but also accelerate from standstill. Consequently, the MPBs of $a_{max}$ can vary greatly depending on the type of action or type of exercise. For example, MPBs are higher in small-sided games, where unpredictable CODs often occur, and lower during constant running drills that lack abrupt changes in direction or speed. Nonetheless, it is worth noting that validity of $a_{max}$ for actions with high accelerations, both from a standstill and from a flying start, were noticeably better when measured with the GPS than with the LPM system.

As recognized above, both measurement systems had the lowest validity of $a_{max}$ when accelerations occurred after an abrupt 180˚ COD (A-COD). During the trial with multiple CODs within 10 m (RA-10m), LPM had higher MPBs in $a_{max}$ than GPS. When the running distance between the CODs was reduced to 5 m (RA-5m), the validity of $a_{max}$ decreased further and was equally poor for both systems (–39.9% for GPS and –42.3% for LPM). As shown in Fig 3, LPM's poor ability to assess abrupt CODs is illustrated by its incomplete deceleration curves prior to the next acceleration: even if the effective speed was 0 for a few tenths of a second during the COD, the measured speed, and therefore also the measured acceleration, never reached 0. This led to a considerably lower validity for LPM $a_{max}$ values when several CODs occurred within short time intervals and without long standstill phases. As mentioned previously, we assume that these incomplete deceleration curves were due to the use of Kalman filters with inadequate predictions [35, 36].

In addition, when acceleration occurred after an immediate deceleration but without a COD (A-D), the validity of $a_{max}$ was noticeably better than in trials with one or more CODs (A-COD, RA-5m, and RA-10m) and was relatively strong for both systems (GPS: 1.1%; LPM: –0.1%). One possible explanation for the large difference in validity of $a_{max}$ between A-D and A-COD, RA-5m, and RA-10m could be due to the Kalman filter settings. In contrast to A-COD, RA-5m, and RA-10m, the direction of A-D is already correctly predicted by the Kalman filter. This leads to the difference in validity between a stop-and-go with COD (A-COD, RA-5m and RA-10m) compared to without COD (A-D). However, this can only be speculated and would have to be verified in further studies with different Kalman filter settings. The chosen test setting allowed only the testing of the two extremes of a stop-and-go action, represented in the subcategories A-D versus A-COD, RA-5m, and RA-10m.

## Validity of speed assessment

We found that the GPS systematically underestimated $v_{max}$; the MPBs of $v_{max}$ ranged from –3.3 to –1.0%. These results are comparable to those of Lacome et al. [18], who found an overall bias of –3.0%. Buchheit et al. [32] observed a positive mean bias of $v_{max}$ of 0.3% for 5 Hz GPS measurements. However, no abrupt 180˚ CODs were performed which produced the highest negative biases in our study. In contrast, the biases of $v_{max}$ measured by the LPM system were unsystematic and ranged from –12.4 to 1.5%. Again, the results of this study are poorer than

the results of Buchheit et al. [32], who measured a mean bias of $v_{max}$ of 0.4% for LPM measurements. Similar to $a_{max}$, validity of $v_{max}$ was markedly lower in actions with multiple CODs than in linear movements: for GPS, the highest bias was –3.3% (RA-5m), while for the LPM system, the bias reached as high as –12.4%. (RA-5m). Again, the main reason for the LPM systems' poor performance in these actions was likely due to the configuration of the Kalman filter, as noted previously [35, 36].

## Inter- and intra-unit reliability

The inter-unit reliability (as measured by CV) for $a_{max}$ was in approximately the same range for both systems (GPS: CV 2.8 to 16.1%; LPM: CV 4.8 to 14.7%). The CV of $a_{max}$ depended on the type of action and was highest for both systems during low accelerations from a standstill. Buchheit et al. [32] reported comparable results for the LPM system (CV ± 10.0% depending on the type of action) but worse results for the GPS (CV >10.0%). The GPS results of inter-unit reliability of this study are similar to those found by Lacome et al. [18], who calculated a CV of 6.4% for $a_{max}$ during a 40 m sprint with a 16 Hz GPS.

The inter-unit reliabilities for both measurement systems were clearly better for $v_{max}$ than for $a_{max}$ and the CVs for $v_{max}$ in every subcategory were within the recommended 5% from Hopkins [38]. The only exception was the RA-5m measured with the LPM system (CV of 5.4%). In contrast, Buchheit et al. [32] found that CV values of $v_{max}$ measured by GPS tend to be higher than those of the LPM system, and are considerably higher than the ones measured in this study. As with $a_{max}$, inter-unit CV measures of $v_{max}$ for GPS are similar to those found by Lacome et al. [18] (CV of 0.5%), but better than those reported in studies by Coutts and Duffield [6] (CV 2.3 to 5.8% for different types of GPS devices) and by Johnston et al. [11] (CV of 8.1%; 15Hz GPS).

Interestingly, inter-unit reliability of $a_{max}$ and $v_{max}$ was generally better than intra-unit reliability for both measurement systems; the only exception to this was $a_{max}$ in A-D with GPS. A possible explanation is that inter-unit reliability was determined simultaneously within the same trial (the RCC carried six devices for each system), whereas the intra-unit reliability was determined by reproducing the same type of action several times throughout the day. Furthermore, satellite availability for the GPS measurements most likely did not remain the same. However, the reason(s) for the low LPM intra-unit reliability are not clear. Different battery charging status is plausible but unlikely. Further research is thus needed to elucidate the factors that influence LPM systems' intra-unit reliability.

## Comparison of local versus global positioning systems

Since LPM systems record with higher frequencies and their base stations are placed around a single site, it was assumed that measurements with the LPM systems would be more accurate than those with GPS. However, this hypothesis has not been confirmed. The validity of $v_{max}$ and $a_{max}$ depended strongly on the types of action and were similar for both the GPS and the LPM system in most of the team sport-specific actions that were analyzed. The only exceptions were found in the HA and RA-10m trials for $v_{max}$, and RA-5m and RA-10m for $a_{max}$ respectively, where GPS surprisingly outperformed the LPM system. The LPM system has significant potential to produce more accurate measurements of speed and acceleration. To benefit from the high potential of the LPM system, it is proposed that sport-specific Kalman filter configurations need to be integrated. Likewise, integrating inertial sensors, such as accelerometers, could allow for further improvements to both GPS and LPM system measurements.

### Study strengths and weaknesses

Due to the use of an RCC, the validity and inter- and intra-unit reliability of $a_{max}$ and $v_{max}$ could be determined without any interference from upper body movements, which have previously been shown to negatively affect the accuracy of laser measurements [16, 28–30]. Additionally, using a vehicle as a device carrier guaranteed laser measurements over a full soccer field without risking a loss of signal. Since the RCC runs on the steel ropes and a platform is attached on top of the RCC, six GPS and six LPM devices can record position measurements simultaneously and without obstructions in a horizontal position at shoulder height, simulating a player wearing the devices. This allowed us to compare the inter-unit reliability of multiple devices in the same trial. However, since the RCC mimicked team sport-specific actions on tensioned steel ropes, not all possible team sport-specific actions could be validated (e.g., acceleration after a COD with an angle of < 180˚). However, all actions included in this study were very well recorded by a laser measurement setup.

The GPS and LPM systems' biases were assessed in comparison to the laser system for each trial, and the results were then averaged to calculate the percentage bias. Thus, the laser system's measurements were compared to GPS and the LPM system measurements for each trial, in order to avoid differences between trials caused by variation in the RCC's operation, rather than the different measurement systems.

We are aware that the technology of position measurement systems is developing extremely fast and the recording frequency of GPS devices has been significantly increased in recent years (from 1 Hz to currently 18 Hz). Although position measurement systems with higher recording frequencies than the ones we have tested now exist, we suppose that the main results of this study can be nevertheless transferred to these systems. This is due to previous results showing that an increase in recording frequency did not automatically result in better data quality [11]. Measurement inaccuracies are likely due to the (wrong) filter configurations, rather than low recording frequencies. Furthermore, the LPM system does not make use of their integrated inertial sensors, although that might greatly improve the measurement accuracy, as they facilitate the differentiation and interpretation of various motor actions. We are convinced that these changes (appropriate filter settings, use of inertial sensors) would improve measurement accuracy to a greater extent than simply increasing the recording frequency, which currently seems to be the main target of the manufacturers.

### Conclusion

Although validity of $v_{max}$ for both systems was generally good (except for LPM measurements during trials with CODs), validity of $a_{max}$ was very low when measured by GPS, and, surprisingly, not higher when measured by the LPM system. The accuracy of $a_{max}$ depended on the type of action, and was considerably lower when actions included any CODs. The Kalman filter configurations are the most likely explanation for this inaccuracy. Given our current knowledge of tracking system accuracy, it is debatable whether acceleration measurements should be included in game or training analyses.

However, acceleration measurements are fundamental to adequately describe physical loads in team sports, and improvements to Kalman filter properties are therefore necessary if team sports with different accelerations and CODs are to benefit from position-tracking systems. In our opinion, using the tested systems in team sports without filter improvements to measure accelerations during CODs is worthless due to the large measurement errors. Nevertheless, if teams do choose to include acceleration as an indicator of physical load, despite the recognized limitations in measuring it, it is essential that they interpret their results with caution.

## Supporting information

**S1 Data.**
(XLSX)

## Acknowledgments

The authors would like to thank Andreas Christen for remodeling the RCC so that it could be used in this study and for serving as the RCC's driver. We would also like to thank Severin Trösch for his valuable support with statistics.

## Author Contributions

**Conceptualization:** Karin Fischer-Sonderegger, Wolfgang Taube, Martin Rumo, Markus Tschopp.

**Data curation:** Karin Fischer-Sonderegger.

**Formal analysis:** Karin Fischer-Sonderegger, Markus Tschopp.

**Investigation:** Karin Fischer-Sonderegger, Martin Rumo, Markus Tschopp.

**Methodology:** Karin Fischer-Sonderegger, Wolfgang Taube, Markus Tschopp.

**Project administration:** Karin Fischer-Sonderegger.

**Resources:** Karin Fischer-Sonderegger, Markus Tschopp.

**Software:** Karin Fischer-Sonderegger, Martin Rumo.

**Supervision:** Wolfgang Taube.

**Validation:** Karin Fischer-Sonderegger, Markus Tschopp.

**Visualization:** Karin Fischer-Sonderegger.

**Writing – original draft:** Karin Fischer-Sonderegger.

**Writing – review & editing:** Karin Fischer-Sonderegger, Wolfgang Taube, Martin Rumo, Markus Tschopp.

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
