## [Decision Letter · Decision Letter 0]

19 Aug 2020

PONE-D-20-09964

How far from the gold standard? Comparing the accuracy of a local position measurement (LPM) system and a 15 Hz GPS to a laser for measuring acceleration and running speed during team sports

PLOS ONE

Dear Dr. Fischer-Sonderegger,

Thank you for submitting your manuscript to PLOS ONE. After careful consideration, we feel that it has merit but does not fully meet PLOS ONE’s publication criteria as it currently stands. Therefore, we invite you to submit a revised version of the manuscript that addresses the points raised during the review process.

We look forward to receiving your revised manuscript.

Kind regards,

Cristina Cortis, Ph.D.

Academic Editor

PLOS ONE

Journal Requirements:

Reviewers' comments:

Reviewer's Responses to Questions

**Comments to the Author**

1. Is the manuscript technically sound, and do the data support the conclusions?

Reviewer #1: Yes

Reviewer #2: Partly

2. Has the statistical analysis been performed appropriately and rigorously? 

Reviewer #1: Yes

Reviewer #2: Yes

3. Have the authors made all data underlying the findings in their manuscript fully available?

Reviewer #1: Yes

Reviewer #2: Yes

4. Is the manuscript presented in an intelligible fashion and written in standard English?

Reviewer #1: Yes

Reviewer #2: Yes

5. Review Comments to the Author

Reviewer #1: Introduction

The introduction is quite long and needs some adjustment. Overall, authors should consider that introduction should lead the reader to understand the current literature on the selected topic and the rationale for the study. In this section as it stands there are many indications about the methods used in the study as well as the what it has been done in this study. Please re-consider this part taking also into considerations the following suggestions:

Line 49: it is necessary referencing the first sentence

Line 51-62: Also here are present too many general information that should be sustained by the use of referenes

Line 74: I would suggest avoiding the use of questions which is more colloquial and not fully suitable in a scientific paper

Line 89-95: this part should be deleted from introduction since it applies to method section

Line 177: “Our investigation is the first to…” This applies to a possible discussion section rather than introduction.

Methods

Line 138: why different number of trials were assessed for each sub-category? Please specify.

Table 1: how the categories were identified? For instance, based on what an acceleration was considered low medium or high? Much more information is required about this categories’ selection and in general about the methods applied. For instance, on which distances were measured, which accelerations and decelerations were adopted etc.

Results

Line 210: abbreviations should be expressed in the text and not with reference to table. Please change accordingly.

Discussion

This section is well-written and easy to follow. I would just suggest to include the limitations of the study and some practical applications for the use of GPS and LPS.

Reviewer #2: General Comments:

The current paper investigated the validity, inter and intra-unit reliability of Local and Global position measurement systems compared to laser measurements. The paper is generally, well written with few editorial concerns (see Specific Comments). However, there is a major problem in why the authors used a filtering technique that they acknowledge was most likely inappropriate for the data (see Specific Comments). Furthermore, in this reviewer’s opinion, they should have been more forceful in their Conclusion by stating the unsuitability of using these devices for determining acceleration during change of direction. As this type of movement is extremely common in sport (as noted by the authors) and the error so large, the methodology is of virtually no use.

Specific Comments:

Line 31-35: Somewhat a run on sentence, which I would suggest breaking apart as follows: "Mean percentage biases (MPBs) of maximal acceleration (amax) and maximal running speed (vmax) were used to measure validity, and mean between-device. Mean within-device standard deviations of the percentage biases (bd-SD and wd-SD) of amax and vmax were used to measure inter- and intra-unit reliability, respectively."

Lines 92-95: What about making the car turn? Although it would not be an exact diagonal movement, it could simulate that sort of change. Even if this is an inappropriate type of movement (i.e. rounding off the direction), the authors should probably address it.

Line 111: Suggest inserting "as a device carrier."

Line 208: The term "MPB" is not defined until Line 225-226. While "MPBs" is defined in the abstract, it should be here as well. Please correct.

Line 210: Please capitalize "Table".

Table 2: Nearly all of the CVs are above 5%, a level that Hopkins suggests renders the performance near useless when considering repeatability (http://www.sportsci.org/resource/stats/index.html). Therefore how practical is it to use these devices for determining acceleration? As that is one of the objectives of the current study, how can the authors suggest the Methods are valid/reliable? (Please see comment in the Conclusion)

Line 222: Is this sentence supposed to be a footnote to the Table? If so, I'd suggest indenting it. Suggest doing the same for Line 254.

Lines 232-234: This reviewer would suggest a bit more explanation as to the "combination" referred to here. I understand that it appears to just be summing them, but why is this useful. An explanation would make it more clear to the reader.

Lines 246-247: Do the authors have any explanation for this variation? There doesn't appear to be anything noted in the Discussion.

Lines 312-316: As the authors were aware of this limitation, why not try a different type of filter? Would it not be possible to use the raw data and do the filtering outside of the device if the filter system was within the device? If this was impossible, then the authors should state why this approach was not used.

Lines 339-340: Please provide some explanation as to why you would make this assumption.

Lines 365-367: While the GPS CVs were almost 10 times lower than LPM, is this really a major issue? Both are below the values recommended by Hopkins for CV (except for RA-5, which is just over 5%), so maybe it doesn't really matter...

Lines 435-436: I would go even further than this caveat. The method should NOT be used for COD, where the error was so high as well as lagging behind.

6. PLOS authors have the option to publish the peer review history of their article (what does this mean?). If published, this will include your full peer review and any attached files.

Reviewer #1: No

Reviewer #2: No

---

## [Author Response · Author response to Decision Letter 0]

17 Feb 2021

Please see separate file labeled "Response to Reviewers".

---

## [Decision Letter · Decision Letter 1]

2 Mar 2021

PONE-D-20-09964R1

How far from the gold standard? Comparing the accuracy of a local position measurement (LPM) system and a 15 Hz GPS to a laser for measuring acceleration and running speed during team sports

PLOS ONE

Dear Dr. Fischer-Sonderegger,

Thank you for submitting your manuscript to PLOS ONE. After careful consideration, we feel that it has merit but does not fully meet PLOS ONE’s publication criteria as it currently stands. Therefore, we invite you to submit a revised version of the manuscript that addresses the points raised during the review process.

As you can see from the reviews' comments, the manuscript vastly improved. There are only few minor comments to be addressed before accepting the paper for publication.

We look forward to receiving your revised manuscript.

Kind regards,

Cristina Cortis, Ph.D.

Academic Editor

PLOS ONE

Journal Requirements:

Reviewers' comments:

Reviewer's Responses to Questions

**Comments to the Author**

1. If the authors have adequately addressed your comments raised in a previous round of review and you feel that this manuscript is now acceptable for publication, you may indicate that here to bypass the “Comments to the Author” section, enter your conflict of interest statement in the “Confidential to Editor” section, and submit your "Accept" recommendation.

Reviewer #1: All comments have been addressed

Reviewer #2: (No Response)

2. Is the manuscript technically sound, and do the data support the conclusions?

Reviewer #1: Yes

Reviewer #2: Yes

3. Has the statistical analysis been performed appropriately and rigorously? 

Reviewer #1: Yes

Reviewer #2: Yes

4. Have the authors made all data underlying the findings in their manuscript fully available?

Reviewer #1: Yes

Reviewer #2: Yes

5. Is the manuscript presented in an intelligible fashion and written in standard English?

Reviewer #1: Yes

Reviewer #2: Yes

6. Review Comments to the Author

Reviewer #1: The authors addressed all my previous comments improving the quality of the manuscript. I would just suggest the addition of this reference, which would be fully suitable in their manuscript:

Conte, D. (2020). Validity of local positioning systems to measure external load in sport settings: a brief review. Human Movement, 21(1).

Reviewer #2: General Comments:

The current version of the manuscript is much improved. There remain a few editorial issues (see Specific Comments), but otherwise only one area of content that should be addressed. I would encourage the authors to provide some explanation for why the validity of acceleration was so different when there were multiple accelerations (see Specific Comments). While they have previously stated their reluctance to speculate on outcomes (an admirable trait), they could still provide a possible explanation, which would provide readers with a basis for what should be examined further.

Specific Comments:

Lines 112-113: Single paragraph sentences should be avoided, if possible. That could be accomplished easily by just combining these two paragraphs. The second is really just carrying on from the first anyway.

Line 127: Suggest replacing "differed" with "differing".

Line 154: Maybe insert "by the device's software". This would assist in indicating that you used the software of the device manufacturer, and didn't filter the data any further.

Lines 169-170: I still have a problem with "linear and multi-directional" being used to describe the movements. Although the direction changed 180 degrees, it was still linear to my way of thinking; as opposed to the "turning" aspect mentioned in the first review. Can the authors come up with a different word for "multi-directional"? I'd encourage that if it were possible. Maybe "forward and backward" or something like that...

Lines 223-229: It is not clear why there are bolded phrases in these lines. If they are to indicate where the figures go, those lines should be separated out from the text, as was done for Figure 1. Otherwise, this section is awkward and difficult to follow what the authors are getting at here.

Lines 233-236: These lines appear to be combined sentences that don't match up. The version in the tracked changes seems to read correct, but this one does not. I think the words as follows need to be removed: "The summarized reliability illustrates the percentage measurement error if a GPS or LPM device is randomly chosen for repeated measures."

Line 318: Typo "smoothed"

Lines 342 & 344: Suggest inserting a comma after "RA-5m". It helps to clarify that you are referring to how A-D differs from all three of the other conditions "A-COD, RA-5m, and RA-10m".

Lines 344-346: Could the authors suggest a possible reason as to why this difference may have occurred? Perhaps it was due to the multiple accelerations that took place in the A-COD, RA-5m, and RA-10m. This is somewhat alluded to in lines 340-342, where the authors state "In addition, when acceleration occurred after an immediate deceleration but without a COD (A-D), the validity of amax was noticeably better than in trials with one or more CODs (A-COD, RA-5m and RA-10m)". Is it possible that the multiple accelerations were "averaged" or the effect was additive? While the authors may not have data to explicitly state this, they could note that additional work could be done to examine this possibility.

Line 406: Suggest replacing "on" with "at".

7. PLOS authors have the option to publish the peer review history of their article (what does this mean?). If published, this will include your full peer review and any attached files.

Reviewer #1: No

Reviewer #2: No

---

## [Author Response · Author response to Decision Letter 1]

31 Mar 2021

Please see "Response to reviewers" letter for specific comments.

---

## [Decision Letter · Decision Letter 2]

12 Apr 2021

How far from the gold standard? Comparing the accuracy of a local position measurement (LPM) system and a 15 Hz GPS to a laser for measuring acceleration and running speed during team sports

PONE-D-20-09964R2

Dear Dr. Fischer-Sonderegger,

We’re pleased to inform you that your manuscript has been judged scientifically suitable for publication and will be formally accepted for publication once it meets all outstanding technical requirements.

Kind regards,

Cristina Cortis, Ph.D.

Academic Editor

PLOS ONE

Additional Editor Comments:

The authors succesfully addressed all the comments and suggestions from the reviewer, and I think the paper can be now accepted for publication.

Reviewers' comments:

Reviewer's Responses to Questions

**Comments to the Author**

1. If the authors have adequately addressed your comments raised in a previous round of review and you feel that this manuscript is now acceptable for publication, you may indicate that here to bypass the “Comments to the Author” section, enter your conflict of interest statement in the “Confidential to Editor” section, and submit your "Accept" recommendation.

Reviewer #1: All comments have been addressed

Reviewer #2: All comments have been addressed

2. Is the manuscript technically sound, and do the data support the conclusions?

Reviewer #1: Yes

Reviewer #2: Yes

3. Has the statistical analysis been performed appropriately and rigorously? 

Reviewer #1: Yes

Reviewer #2: Yes

4. Have the authors made all data underlying the findings in their manuscript fully available?

Reviewer #1: Yes

Reviewer #2: Yes

5. Is the manuscript presented in an intelligible fashion and written in standard English?

Reviewer #1: Yes

Reviewer #2: Yes

6. Review Comments to the Author

Reviewer #1: No further comments. The authors addressed all the comments and the manuscript reached a level suitable for publication.

Reviewer #2: (No Response)

7. PLOS authors have the option to publish the peer review history of their article (what does this mean?). If published, this will include your full peer review and any attached files.

Reviewer #1: No

Reviewer #2: No

---

## [Editor Report · Acceptance letter]

15 Apr 2021

PONE-D-20-09964R2 

How far from the gold standard? Comparing the accuracy of a local position measurement (LPM) system and a 15 Hz GPS to a laser for measuring acceleration and running speed during team sports 

Dear Dr. Fischer-Sonderegger:

I'm pleased to inform you that your manuscript has been deemed suitable for publication in PLOS ONE. Congratulations! Your manuscript is now with our production department. 

Kind regards, 

on behalf of

Prof. Dr. Cristina Cortis 

Academic Editor

PLOS ONE